# Interventions promoting recovery from depression for patients transitioning from outpatient mental health services to primary care: A scoping review

Anne Sofie Aggestrup[1⊙]*, Frederik Martiny[2,3⊙], Line Lund Henriksen[1], Annette Sofie Davidsen[2⊙], Klaus Martiny[1⊙]

**1** The Research Unit for Mental Health Centre Copenhagen, Copenhagen Affective Disorder Research Centre (CADIC), New Interventions in Depression (NID) Group, Mental Health Services in the Capital Region of Denmark, University of Copenhagen, Frederiksberg Hospital, Frederiksberg, Denmark, **2** The Research Unit for and Section of General Practice, Department of Public Health, University of Copenhagen, Copenhagen, Denmark, **3** Center for Social Medicine, Bispebjerg and Frederiksberg Hospital, Frederiksberg, Denmark

⊙ These authors contributed equally to this work.
* anne.sofie.aggestrup@regionh.dk

## Abstract

### Introduction

Major Depressive Disorder (MDD) is one of the most prevalent mental disorders worldwide with significant personal and public health consequences. After an episode of MDD, the likelihood of relapse is high. Therefore, there is a need for interventions that prevent relapse of depression when outpatient mental health care treatment has ended. This scoping review aimed to systematically map the evidence and identify knowledge gaps in interventions that aimed to promote recovery from MDD for patients transitioning from outpatient mental health services to primary care.

### Materials and methods

We followed the guidance by Joanna Briggs Institute in tandem with the PRISMA extension for Scoping Reviews checklist. Four electronic databases were systematically searched using controlled index–or thesaurus terms and free text terms, as well as backward and forward citation tracking of included studies. The search strategy was based on the identification of any type of intervention, whether simple, multicomponent, or complex. Three authors independently screened for eligibility and extracted data.

### Results

18 studies were included for review. The studies had high heterogeneity in design, methods, sample size, recovery rating scales, and type of interventions. All studies used several elements in their interventions; however, the majority used cognitive behavioural therapy conducted in outpatient mental health services. No studies addressed the transitioning phase

**Data Availability Statement:** All data generated or analyzed during this scoping review is available from OSF at https://osf.io/5894b/, and from the cited studies and the Supporting Information files.

**Funding:** The study was funded by Helsefonden (21-B-0478), Jaschafonden (2021-0082) and Tværspuljen (P-2022-1-08 and P-2023-1-15). The funding source supports the investigators salary to carry out this review. The funders were not involved in the study design, collection, analysis, or interpretation of data, writing the manuscript, and the decision to submit the manuscript for publication.

**Competing interests:** The authors have declared that no competing interests exist.

from outpatient mental health services to primary care. Most studies included patients during their outpatient mental health care treatment of MDD.

## Conclusions

We identified several knowledge gaps. Recovery interventions for patients with MDD transitioning from outpatient mental health services to primary care are understudied. No studies addressed interventions in this transitioning phase or the patient's experience of the transitioning process. Research is needed to bridge this gap, both regarding interventions for patients transitioning from secondary to primary care, and patients' and health care professionals' experiences of the interventions and of what promotes recovery.

## Registration

A protocol was prepared in advance and registered in Open Science Framework (https://osf.io/ah3sv), published in the medRxiv server (https://doi.org/10.1101/2022.10.06.22280499) and in PLOS ONE (https://doi.org/10.1371/journal.pone.0291559).

## Introduction

### The global health burden of depression

Major Depressive Disorder (MDD) is one of the most prevalent mental disorders globally [1–4], affecting an estimated 350 million people worldwide [5]. The disorder has significant personal and public health consequences [6, 7], and a major impact on society, including direct and indirect costs [8, 9]. Following a person's first episode of MDD (S1 Appendix), the risk of relapse is more than 50%, rising to 70% and 90% following a second and third episode respectively [10–18]. The risk of hospitalization increases following each new episode of MDD [9, 12, 19–21]. Thus, MDD is regarded as a chronic recurrent condition that requires long-term management and recovery maintenance to prevent relapse [10, 22, 23]. Patients with mental disorders are mainly treated in primary care [4, 24]. In any given 12-month period, 10–20% of adults will visit their general practitioner (GP) for mental complaints, most of them related to depression [25], and when ending treatment in the mental health services, the responsibility for the continued follow-up lies within primary care.

### Treatment and management of depression

Research investigating interventions that can promote recovery has primarily been conducted in a primary care setting. Cochrane reviews [26, 27] showed that shared care with mental health services improves depression outcomes in primary care. Another study found that a multifaceted intervention consisting of collaborative management by the primary care physician and a consulting psychiatrist, intensive patient education, and surveillance of continued re-prescriptions of antidepressant medication improved adherence to antidepressant regimens in patients with both major and minor depression [28–30]. To our knowledge, no studies have investigated collaborative care for relapse prevention for patients with MDD who have been discharged from outpatient mental health services (i.e., outpatient hospital-based mental health services, S1 Appendix) and continue their treatment in primary care.

## How to measure treatment effects on depression - Clinical and personal recovery

The concept of recovery has become incsreasingly important among stakeholders [31]. Treatment success in mental health services for people with MDD is perceived as progress in terms of the degree of recovery. However, recovery is a complex concept, and the meaning of the concept has changed over time, resulting in heterogeneous definitions of the concept [32–34]. Authors define clinical recovery as comprising partial or full symptom remission, independent living, gaining control over the illness, and full or part-time work or education [34–36]. Personal recovery, having emerged from the mental health service user movement, refers to a process in which the individual recovers from the social consequences of mental illness and regains a meaningful life, participating in the community and overcoming the challenges of mental illness with or without symptoms [37–39]. A recent systematic review and narrative synthesis of personal recovery has received attention as a way to operationalize personal recovery [40]. The synthesis resulted in a conceptual framework: Connectedness; hope and optimism about the future; identity; meaning in life; and empowerment (giving the acronym CHIME) [40]. The importance of CHIME is widely endorsed in the literature [41], which makes the framework suitable for evaluating personal recovery. Thus, 'personal recovery' is more than a reduction of symptoms. It is the subjective experience of having mental health difficulties [42–45]. However, it is argued that the two terms should be considered complementary rather than contrasting [46]. In the current review, we focus on both clinical and personal recovery.

## The process of recovering from depression

Research shows that most patients with MDD do not achieve symptomatic remission or full recovery after treatment in outpatient mental health services [47–49]. Therefore, they are at risk of developing new episodes of MDD or treatment-resistant depression [50]. Mental health services are challenged in the attempt to promote recovery-oriented interventions due to sparse in- and outpatient resources [51, 52]. Treatment typically focuses on providing fast diagnostic assessment and medical stabilization with a focus on clinical recovery [53, 54]. However, attempts to promote personal recovery in outpatient mental health services before discharge to primary care have focused on various strategies, e.g., psychoeducation [55], pharmacological treatment, Basic Body Awareness Therapy, a variety of physical elements, and complementary medicine [56], all provided by a team of mental health care professionals. In a single-arm feasibility trial using electronic self-monitoring authors significantly drift the sleep-wake cycle associated with mood worsening after discharge from an inpatient mental health care treatment [57]. In a subsequent randomized controlled trial (RCT) using a circadian reinforcement therapy (CRT) intervention supported by an electronic self-monitoring system authors found significantly lower depression levels, improved sleep quality, lower day-to-day variability in daily sleep, mood parameters, and activity parameters in the intervention group than in the control group after discharge from treatment in outpatient mental health care [58, 59]. A review from 2021 [60] summarized categories of common transition practices when stable patients transitioned from outpatient mental health services to primary care. The review identified a profound need for more research in the area [60]. To our knowledge, no reviews have studied interventions aiming at promoting recovery of patients transitioning from secondary to primary care, and whether they prevent relapse of MDD and readmissions to inpatient mental health services.

### Aim and research questions

This scoping review aimed to systematically map the evidence and identify knowledge gaps in studies of interventions that aimed to promote recovery from MDD for patients transitioning from outpatient mental health services to primary care.

We posed the following four research questions (RQs):

*RQ1*: *What characterizes the studies in terms of settings, aims, and methods*?

*RQ2*: *How do studies define and measure recovery*?

*RQ3*: *What are the intervention content and implementation strategies*?

*RQ4*: *What are the findings of the studies as to what promotes recovery from MDD in patients transitioning from outpatient mental health services to primary care*?

## Materials and methods

We chose to conduct a scoping review since this type of review is well-suited to map the evidence and identify knowledge gaps when little is known about the topic and when the knowledge is of a complex or heterogeneous nature [61]. Scoping reviews are valuable for systematically synthesizing broad-based evidence on intersectoral and interdisciplinary relevance, building bridges, and bringing coherence to a diverse evidence base [62–67].

This review followed the methodological guidance from the Joanna Briggs Institute (JBI) [68, 69] in tandem with the Preferred Reporting Items for Systematic Reviews and Meta-Analysis - extension for Scoping Reviews (PRISMA-ScR) checklist [62] (S2 Appendix).

### Protocol and registration

A protocol was prepared in advance and registered in Open Science Framework (OSF) (https://osf.io/ah3sv), published in the medRxiv server (https://doi.org/10.1101/2022.10.06.22280499) and in PLOS ONE [70]. All data generated or analysed during this scoping review is available online: https://osf.io/5894b/ from the cited studies that we included in this published article, and its supplementary information files.

### Eligibility criteria

The eligibility criteria were developed from the research questions by the multidisciplinary review group. The eligibility criteria were structured according to the 'PICOS' acronym (Population, Intervention, Comparator, Outcome, and Setting) (S3 Appendix). Although the outcome category in PICOS was not part of the criteria for including qualitative articles, the acronym helped to specify the search strategy to answer both the qualitative and qualitative RQs.

### Information sources

Literature was searched in the scientific electronic databases of Medline via PubMed, PsycINFO, CINAHL, and Sociological Abstracts. The search strategy included both text words and Medical Subject Headings (MeSh) / Thesaurus headings terms. The primary search was performed between the 20th of January and the 29th of March 2022. An updated search in all databases was conducted on the 13th of October 2023. Reference lists of included articles were hand-searched for eligible articles and backward and forward citation tracking was performed on included articles.

## Search

The literature search was developed in collaboration with an information specialist with feedback from the stakeholders as described in the TRANSFER approach [71]. The search strategy for PubMed is available in the S4 Appendix. Before performing the search strategy, we searched for ongoing or completed scoping or systematic reviews in the area in the Cochrane Library, Google Scholar, and the PROSPERO register to make sure that no similar reviews had been published.

## Selection of sources of evidence

Records from databases were transferred to EndNote 20 for the removal of duplicates [72]. Subsequently, all unique records were uploaded to Covidence [73]. Three review authors (ASA, LLH, and LF) independently screened on the title and abstract level. Records that were included on the title-abstract level were subsequently read in full text. Disagreements were discussed in the team in physical or virtual meetings until a consensus was reached. A fourth reviewer (KM) resolved further conflicts regarding quantitative articles, and a fifth reviewer (ASD) resolved further conflicts regarding qualitative articles.

## Data charting process

Data from the included articles were extracted independently by the review team (ASA, LLH, and LF) using a data extraction form developed in a Microsoft Excel sheet by ASA and FM inspired by the JBI guideline [68, 69] and the Template for Intervention Description and Replication (TIDieR) checklist [74].

## Data items

Data was extracted regarding study characteristics (authors, title, year of publication, journal, country, and study design), participants (population number, gender, and age), methodology, intervention characteristics, setting of intervention(s), key findings relating effects of the intervention(s), and facilitators/barriers for recovery.

## Critical appraisal of individual sources of evidence

Since this is a scoping review, we have not conducted a quality appraisal, which is consistent with the framework proposed by the JBI methodology for scoping reviews [66, 75].

## Synthesis of results

The qualitative data was analysed to produce a descriptive synthesis of results. Quantitative data was summarized in tables. Mixed methods data was tabulated and summarised.

## Patient and public involvement

We used a partnership approach within the TRANSFER approach [71] to complete this review. Our review team contained researchers with academic, primary health care, and secondary health care experience, including social medicine, mental health services, and general practice, bringing a range of different skills and perspectives.

## Ethics

This scoping review constitutes the first step in a larger research project aiming to develop a complex intervention to promote recovery and prevent relapse of MDD when the treatment is

transferred outpatient mental health services to primary care. The methodology is based on publicly available information and does not require ethical approval.

## Results

### Selection of sources of evidence

For the systematic search, 5070 studies were identified, of which 202 full texts were assessed for eligibility (Fig 1). Of these, 186 were excluded, primarily due to the wrong setting ($n = 64$), as most studies were conducted in primary care without a reported prior treatment in mental health services, or wrong outcome ($n = 34$), i.e., not reporting on recovery. Following full-text reading, 16 studies met eligibility criteria and were included in the synthesis. After backward and forward tracking of studies included for review two additional studies were included. In total, 18 studies formed the final dataset and proceeded to data extraction [76].

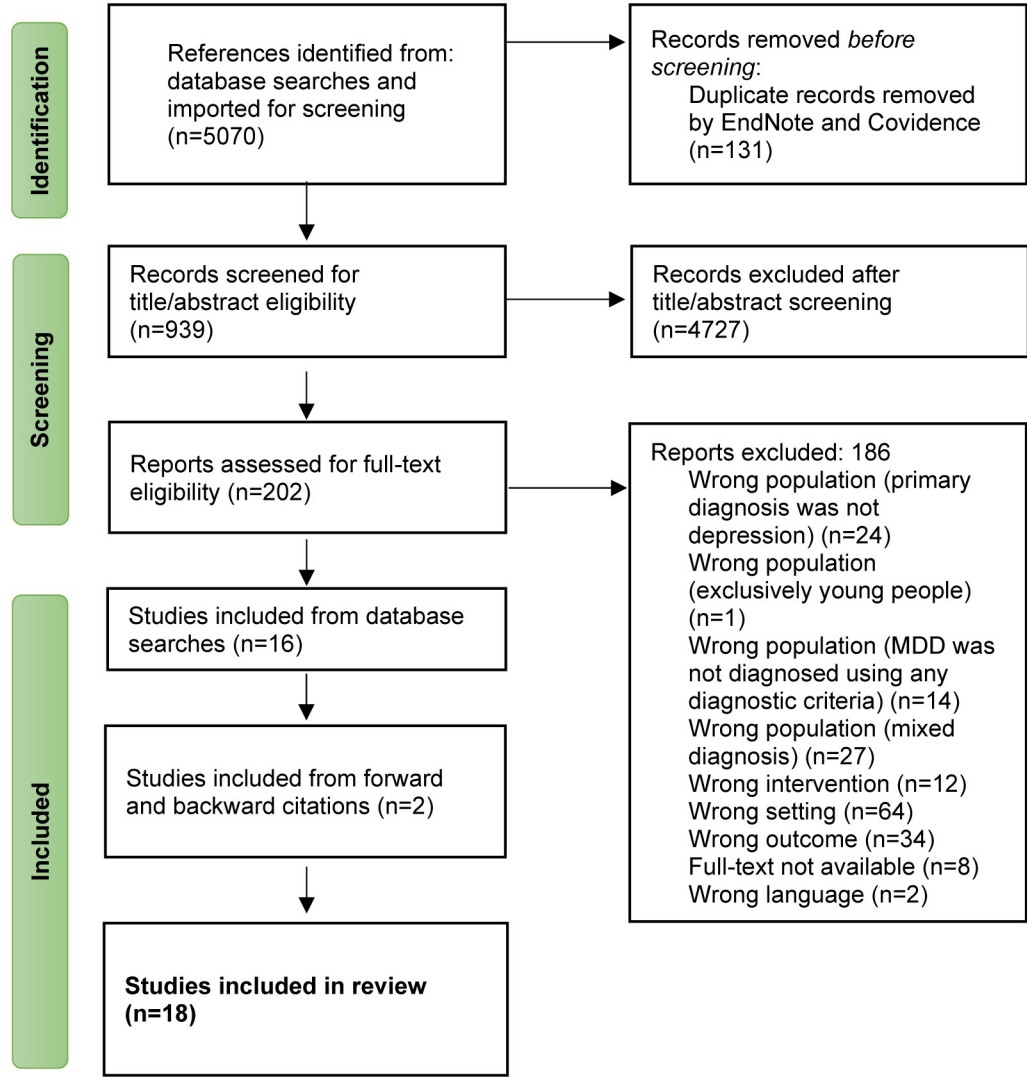

**Fig 1. PRISMA-ScR flow diagram for the study selection.**

## Characteristics of sources of evidence

**RQ1: What characterizes the studies in terms of settings, aims, and methods?.** Information about authors, year of publication, aim, country and setting, recruitment, study period, method, and research design are summarized (Table 1). Included studies were published between 1996 and 2023, with most studies published within the last 10 years (Fig 2). All studies were conducted in high-income, western countries, including the United States of America (USA) (*n* = 5), Sweden (*n* = 4), Australia (*n* = 2), Canada (*n* = 2), Denmark (*n* = 3), Norway (*n* = 1) and Spain (*n* = 1) (Fig 3).

Most studies (*n* = 13) used quantitative methodologies, one used a mixed methods approach, and four studies used qualitative methods. The quantitative and mixed methods studies were RCTs (*n* = 8), consisting of two arms (*n* = 6; [77–82]) or three arms (*n* = 2; [83, 84]), non-randomized pre-post trials (*n* = 1; [85]), prospective studies (*n* = 3; [86–88]), retrospective studies (*n* = 1; [89]), and single-arm intervention trials (*n* = 1; [90]). The quantitative studies collected data either via assessor rating questionnaires, patient-reported questionnaires, or from patient records. The qualitative studies [82, 91–93] collected data through individual interviews, and the mixed-method study [86] collected data via assessor rating questionnaires and individual interviews. It varied whether participants had a first-time episode of MDD or recurrent MDD. Most of the included studies collected data in mental health services (*n* = 14).

Participants in the studies had a diagnosis of MDD according to the criteria of the *Diagnostic and Statistical Manual of Mental Disorders*, Fourth Edition (*DSM*-IV) [100] or *the International Statistical Classification of Diseases and Related Health Problems 10$^{th}$ Revision* (ICD-10) [101]. Characteristics of study populations varied across studies (S5 Appendix). In total, 2821 participants participated across studies (mean, *n* = 156.7), with 1915 participants (mean, *n* = 106.4) completing data collections (67.9%). The age of participants ranged from 18 to 69 years. The mean age range was 33.1 years– 53.8 years, with a female percentage range from 44.4% to 100%. Nine studies reported psychiatric comorbidity in addition to MDD. e.g., anxiety, phobias, or personality disorders.

The sample size was reported in four different ways across studies; *1) whole sample (W)*: The number of participants in the entire experimental group from the beginning of the study; 2) *baseline sample (B)*: The number of participants included in the study before randomization; *3) randomized sample (R)*: After random allocation to an experimental group, and *4) completed sample (C)*: The number of participants completing the experiment.

## Results of individual sources of evidence

**RQ2: How do studies define and measure recovery?.** Overall, quantitative studies primarily focused on *clinical recovery*, whereas qualitative studies focused more on *personal recovery*. Quantitative studies (when reported) used vague descriptions of clinical recovery focusing on clinical change and symptoms, such as *'clinically significant improvement'* [85], *clinically significant change'* [78], and *'few or no symptoms consistently for extended periods'* [81], without a definition of the recovery concept. All qualitative studies and two of the quantitative studies [80, 87], included elements of recovery focusing on the individual's own process and support from their social network in their definitions, e.g., *well-being, support from others, hope, goals, self-esteem, positive attitude, and gaining knowledge*. They used the same terms, e.g., *hope, well-being, and that recovery is a nonlinear ongoing process*, words deriving from the first definition of personal recovery by William Anthony in 1993 [102].

Only one of the quantitative studies [80] and two of the qualitative studies [82, 91] defined their use of the recovery concept.

Table 1. Characteristics of studies (*n* = 18).

| Author and year | Aim | Country and setting | Recruitment | Study period | Research Design |
|---|---|---|---|---|---|
| | | Quantitative studies (*n* = 13) | | | |
| Callesen et al. (2020) [94] | To assess the clinical efficacy of metacognitive therapy (MCT) compared to current best practice, cognitive behaviour therapy (CBT), in adults with MDD. | An outpatient mental health clinic, Næstved, Denmark | Referred by GPs. | January 2011 - June 2015 | Parallel single-blind randomized controlled trial. |
| Craigie and Nathan (2009) [85] | To evaluate the effectiveness of group CBT compared to individual CBT for depressed outpatients in a naturalistic setting. | Centre for Clinical Interventions, a community-based and government-funded adult outpatient mental health clinic, Perth, Australia. | Referred by professionals in the private and public sector, e.g., psychiatrists and GPs. | 2001–2006 | Nonrandomized pre-post design with no long-term follow-up assessment. |
| Ekeblad et al. (2016) [78] | To compare two treatments, interpersonal psychotherapy (IPT) and CBT by analysing whether IPT was noninferior to CBT in reducing depression. | An outpatient mental health care clinic, Sundsvall, Sweden. | Referred by GPs. | 2010–2013 | Randomized trial with a non-inferiority design. |
| Tønning et al. (2021) [79] | To investigate whether a smartphone-based monitoring and treatment system reduces the rate and duration of readmissions, more than standard treatment, in patients with unipolar depressive disorder following hospitalization. | Mental Health Centre Copenhagen, the Capital Region of Denmark. | Recruited from mental health services. | May 2017 - August 2019. Last visit: 2020. | A pragmatic, parallel-group, rater-blinded randomized controlled trial. |
| Thimm and Antonsen (2014) [89] | To retrospectively evaluate the effectiveness of group CBT for depression administered in an outpatient mental health care clinic. | Mental Health Centre of the Helgeland Hospital Trust, Mo I Rana, Norway. | Referred by GPs or specialized health services. | 2002–2013 | Retrospective study design. |
| Ezquiaga et al. (1998) [88] | Patients with non-chronic MDD were followed up prospectively for 6 months to investigate clinical, social, and cognitive variables. | Four outpatient mental health centres, Madrid, Spain. | Not reported. | 10 months in 1994 | Longitudinal, observational, analytical, and prospective in design. |
| Ludman et al. (2016) [80] | To determine whether a self-management support service was more effective than treatment as usual in reducing depressive symptoms and major depressive episodes and increasing personal recovery among individuals with chronic or recurrent depressive symptoms. | Four primary care clinics, all non-profit health care organizations in the Seattle, Washington area, USA. | Recruited from primary care clinics, identified by computerized records, or referred by professionals. | January 2010 - October 2011 | Randomized controlled trial. |
| Tutty et al. (2010) [90] | To test the feasibility and effectiveness of a stand-alone, eight-session CBT telephone treatment (CBT-TT) program for depression and to benchmark outcomes against a previous phone counseling trial combined with pharmacotherapy. | An outpatient mental health clinic, USA. | Referred by mental health services. | Not Reported. | Single-arm intervention. |
| Vittengl et al. (2010) [81] | To identify demographic, cognitive, social interpersonal, and personality variables to clarify which patients' continuation-phase CT helps to avoid relapse and recurrence and achieve remission and recovery in a randomized clinical trial [95]. | Outpatients, USA. | Not reported. | Not reported. | Randomized controlled trial. |
| Jarrett et al. (2013) [83]* | To test the efficacy of the continuation phase model of Cognitive Therapy (C-CT) and fluoxetine (FLX) for relapse prevention in a placebo (PBO) controlled randomized trial and compare the durability of prophylaxis after discontinuation of treatments. | Outpatients from university-based specialty clinics, USA. | Recruited by clinical referrals and advertisements. | March 2000 - July 2008. Completed follow-up: May 2011. | Randomized controlled trial, including a parallel sequential, three-stage design. |

*(Continued)*

**Table 1.** (Continued)

| Author and year | Aim | Country and setting | Recruitment | Study period | Research Design |
|---|---|---|---|---|---|
| Vittengl et al. (2016) [84]* | To clarify social-interpersonal functioning after response to acute-phase CT for MDD, including durability of improvements in social-interpersonal functioning, effects on continuation treatment, and prediction of depressive symptoms and relapse / recurrence. | Outpatients from university-based specialty clinics, USA. | Recruited by clinical referrals and advertisements. | March 2000 - July 2008. Completed follow-up: May 2011. | Randomized controlled trial, including a sequential, three-stage design. |
| Skärsäter et al. (2005) [87] | To explore the sense of coherence and social support in patients treated in mental health services for a first episode of unipolar major depression. | Ambulatory medical centres and mental health open care units, Sweden. 1-year follow-up interviews in an outpatient Affective Unit. | Recruited from ambulatory medical centers and mental health open care units. | May 1998. Follow-ups every 6 months for 4 years. | Prospective and longitudinal |
| Steig et al. (2023) [96] | To investigating the effect of transdiagnostic group CBT vs. standard diagnosis-specific group CBT for depression, agoraphobia/panic disorder and social anxiety disorder. | Three Danish regional mental health services. | Recruited from one Psychotherapeutic Clinic and two Outpatient Clinics. | 2016–2018. Follow-up: 2017–2019. | A pragmatic, non-inferiority, randomized controlled clinical trial |
| **Mixed method study (n = 1)** | | | | | |
| Lawn et al. (2019) [86] | To report on the first 17 months of the MindStep™ outcome, implemented across Australia from March 2016, in a cohort of clients. MindStep™ forms part of a stepped-care model to address the transition from acute to community mental health care in Australia. | The LiCBT coaches delivered the program in Melbourne and the supervisors were based in Adelaide, Australia. | Referred from private health funds, following an acute hospital admission for anxiety and/or depression. | March 2016 - July 2017. | Prospective observational study with questionnaires and semi-structured interviews analysed by Framework Analysis. |
| **Qualitative studies (n = 4)** | | | | | |
| Woolley et al. (2020) [91] | To understand how recovery-oriented occupational therapy groups shape participants' personal experience of daily life, including recovery. | Outpatients in a mental health service department at a university-affiliated hospital in Montreal, Quebec, Canada. | Recruited from a recovery-oriented outpatient program in a mental health care department. | Summer of 2018. | In-depth semi-structured interviews analysed by Interpretative phenomenology (IPA) [97]. |
| Peden (1996) [92] | To describe the process of recovering in women who have been depressed. | Women who had previously been hospitalized with depression were interviewed in their homes, in Sweden. | Recruited by psychiatric nurses and friends who knew women recovering from depression. | Initially, data were collected in 1991 [98], and the study performed 1-year follow-up interviews. | In-depth follow-up interviews guided by Peplau [99], analysed by practice-based theory development. |
| Skärsäter et al. (2003) [93] | To describe, from a salutogenic approach, women's conceptions of coping with major depression in daily life with the help of professional and lay support. | Swedish-speaking women who had received inpatient care for depression decided on a convenient time and place for an interview. | Women, previously hospitalized for MDD were recruited by the author of the study. | October 1999 - March 2000. | In-depth individual interviews analysed by phenomeno-graphic analysis. |
| Bouchal et al. (2023) [82] | To explore the processes of personal recovery in patients with treatment-resistant depression following deep brain stimulation of the subcallosal cingulate. | Patients treated at an academic hospital in Calgary, Canada. | Recruited from the original cohort of patients who participated in the DBS trial. | Not reported | Semi-structured interviews were analysed by a constructivist grounded theory approach. |

**\*Jarett et al. et al.** (2013) [83] and Vittengl et al. (2016) [84]* used the same dataset

*Definition and measurement of recovery.* All quantitative studies and the mixed method study provided a recovery scale and a cut-off score to define how and when participants had recovered (Table 2). Across quantitative studies and the mixed method study, seven different rating scales were used to measure recovery (S6 Appendix). Of these, only RAS focuses on personal recovery. It was used in two studies [79, 80]. The most common outcome measure was Beck Depression Inventory (BDI-II) (*n* = 5) followed by Psychiatric Rating Scale (PSR) of DSM-IV MDD (*n* = 3), Recovery Assessment Scale (RAS) (*n* = 2), Hamilton Rating Scale for Depression (HAM-D) 17-item (*n* = 1), Symptom Checklist Depression Scale (SCL) 20-item (*n* = 1), Montgomery & Åsberg Depression Rating Scale (MADRS) (*n* = 1), and Patient Health

Questionnaire (PHQ) 9-item (*n* = 1). Most studies used self-reported rating scales (RAS, BDI-II, PHQ-9, SCL-20), and some used clinician's rated scales (HAM-D-17, and MADRS). BDI-II cut-off scores varied between studies.

*Definitions of the recovery concept in qualitative studies (n = 4)*. Woolley et al. (2020) [91], referring to the Mental Health Commission of Canada, defined recovery from mental illness as a condition, where one can live a "*hopeful, satisfying and contributing life, even with ongoing limitations*" [104], p. 16. With reference to Deegan et al. [39], they considered recovery as a nonlinear, ongoing lifelong process, but also an outcome, where individuals continuously learn about their illness and make an active effort to manage it [105].

Peden (1996) [92], using Peplau's process of practice-based theory development [98] described the process of recovery, not the concept, as consisting of eight categories in three phases. The process is described as dynamic and non-linear with movement within and across phases and always with a "turning point" experience.

Skärsäter et al. (2003) [93] used a salutogenic approach to describe women's coping with MDD in daily life placing the ability to cope with depression on equal footing with recovery. They pointed out that there is a conflict between depressed patients' conceptions of their problems and the treatment options and current professional depression management. The women in the study wanted a more holistic view of their needs.

Bouchal et al. (2023) [82] distinguish between clinical and personal recovery. Their focus is on personal recovery, which, concerning Barber [106], means that "*one functions as one's best despite ongoing symptoms, developing new meaning and purpose*" (p. 1007; [82]), and with reference to Noiseux and Ricard [107], the personal recovery process involves "*change to self, embracing hope, optimism, and empowerment*" (p. 1007; [82]). "*Personal recovery from MDD is a complex process that is unique individually and in the family system, undertaken by family and caregivers*" (p. 1007; [82]).

**RQ3: What are the intervention content and implementation strategies?.** *Intervention content*. Almost all studies (*n* = 16) involved an intervention, except two qualitative studies. In one of the qualitative studies, women who had experienced a depressive episode were interviewed about their recovery process [92], and in the other study, women's conceptions of coping with MDD in daily life with help from professional and lay support were explored [93].

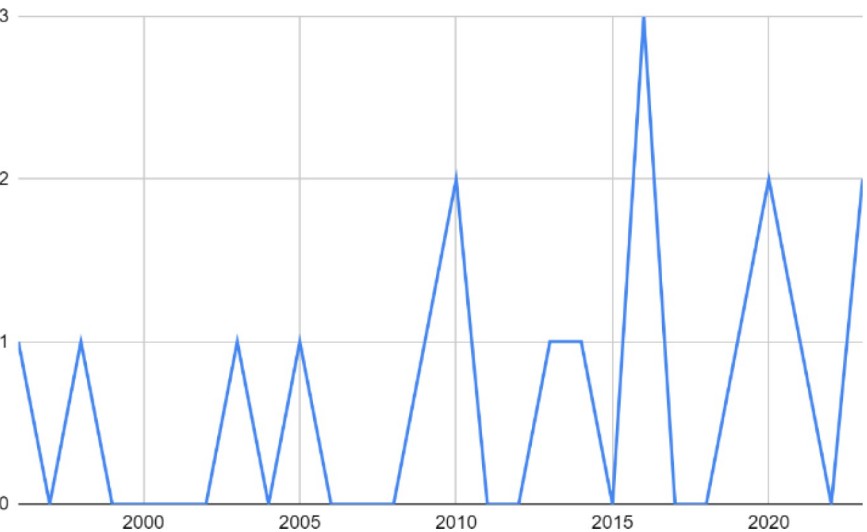

**Fig 2. Number of publications per year.**

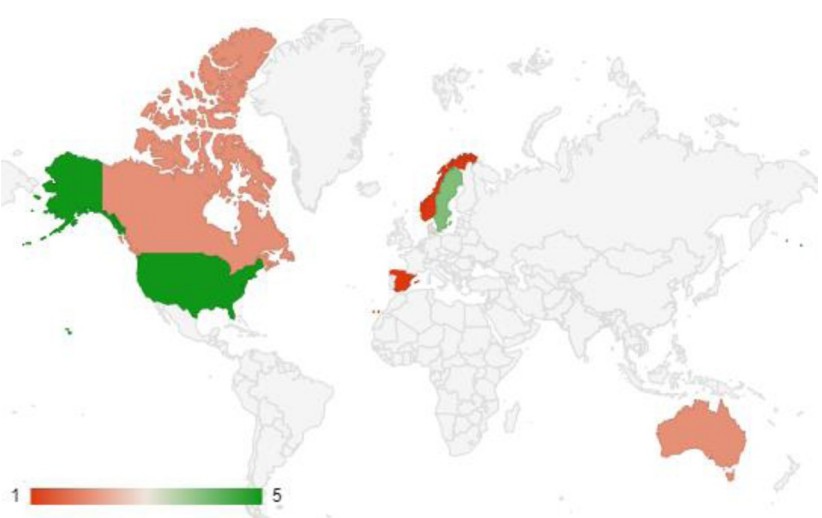

**Fig 3. Publication output.**

**Table 2. Definitions of the recovery concept in quantitative studies (*n* = 14).**

| Author and year | Definition of the recovery concept | Scale | Cut-off score |
|---|---|---|---|
| Callesen et al. (2020) [94] | Not defined. | BDI-II | BDI-II scores below 11 at post-treatment and reduced by 10 points or more from pre-treatment. |
| Craigie and Nathan (2009) [85] | Not defined. | BDI-II | BDI-II scores below 13.5 and a more conservative BDI-II cut-off score on ≤9. |
| Ekeblad et al. (2016) [78] | Not defined. | BDI-II | BDI-II scores below 10. |
| Tønning et al. (2021) [79] | Not defined. | RAS | RAS, item scale and cut-off score not reported. |
| Thimm and Antonsen (2014) [89] | Not defined. | BDI-II | BDI-II scores below 16.66. |
| Ezquiaga et al. (1998) [88] | Not defined. | HAM-D-17 | HAM-D-17 [103] scores below 8. Item scale not reported. |
| Ludman et al. (2016) [80] | A complementary, ongoing processes in which a person is the central determinant of his or her health and well-being. | RAS | RAS, item scale and cut-off score not reported. |
| Tutty et al. (2010) [90] | Not defined. | SCL-20 | Reliable and significant changes in SCL-20 with cut-off scores at 0.5. |
| Vittengl et al. (2010) [81] | Not defined. | PSR | Weekly PSR ratings of DSM-IV MDD (on a 1–6 scale) of 1 (no symptoms) or 2 (one or two mild symptoms) for ≥ 35 for continuous weeks during 24 months of follow-up. |
| Jarett et al. (2013) [83] | Not defined. | PSR | Remission lasting ≥ 8 consecutive months. |
| Vittengl et al. (2016) [84] | Not defined. | PSR | Weekly PSR ratings of DSM-IV MDD (on a 1–6 scale) of 1 (no symptoms) or 2 (one or two mild symptoms) for ≥ 35 for continuous weeks. |
| Skärsäter et al. (2005) [87] | Not defined. | MADRS | MADRS scores below or equal to 8. |
| Steig et al. (2023) [96] | Not defined. | BDI-II | BDI-II scores below 19 at follow-up. |
| Lawn et al. (2019) [86] | Not defined. | PHQ-9 | PHQ-9 > 9. |

The content of the interventions varied considerably across studies, both in terms of the elements, manuals used to guide the intervention, and the length and method of delivery. We identified 17 distinct intervention elements across studies (S7 Appendix); they are further elaborated in S8 Appendix. Many studies followed a published manual to guide their intervention (50%), e.g., a standard published treatment manual from 1979 by Beck and colleagues [108, 109] to guide delivery of CBT. One study [79] used a Smartphone app, *Monsenso* as an alternative to written manuals. Most interventions (*n* = 9) were compared with either active or control comparators. Typically, the content of both the active and the comparator intervention included a combination of three to four different elements, e.g., CBT, psychoeducation, and mindfulness.

*Intervention elements.* Most studies (*n* = 14) included cognitive restructuring and/or challenging negative automatic thoughts to reduce depressive symptoms, e.g., CBT, MCT, Rumination Focused Cognitive Behavioural Therapy (RFCBT) [78–86, 89–91, 94, 96]. Seven studies included elements of psychoeducation, i.e., information about depression [79, 83–85, 89–91]. An increased focus on participants' social support or network was reported in seven studies and included interpersonal therapy [78] interpersonal relationships [89], peer support [80], identifying interpersonal vulnerabilities [83], and social prescribing [83, 84, 86].

In addition, one study [91] focused on supporting participants' return to work/supported employment. Many studies (*n* = 6) focused on planning and activity scheduling, e.g., activity scheduling, mastery and pleasure techniques [80, 85, 86, 89, 91, 94], and troubleshoot actions to meet personal self-management goals [80]. Five studies had elements of homework [85, 94], e.g., workbook exercises [86, 89], or daily patient-reported entries [79]. Three studies used motivational interviewing [90] to make a change in the participants' life [91] or employed evocative and motivational strategies to increase engagement [74]. Two studies used mindfulness and calming techniques [85, 94]. Four studies reported using coping strategies, including how to live with a mental illness [91] and coping with thoughts and emotions [83, 84, 94]. Two studies used elements of recovery to improve patient's enjoyment of life by promoting a sense of well-being, and optimism [80] and elements of recovery were used in an occupational group [91]. Finally, a qualitative study used a medical device in the form of DPS and add-on CBT [82].

*Intervention duration, delivery, and tailoring.* Interventions varied in duration (from eight to 24 sessions) and length (from 6 weeks to 18 months). Most interventions were delivered by psychologists (*n* = 7) and/or psychiatric nurses (*n* = 3). Some psychologists had experience from working in mental health services and/or had received training in providing the intervention. Interventions were delivered individually (*n* = 6), in groups (*n* = 4), by telephone (*n* = 3), pharmacologically (*n* = 3), or integrated into a smartphone app including clinical feedback (*n* = 1). In addition, one study [85] investigated different ways of delivering CBT, individually or in groups. Most of the studies (*n* = 13) tailored their interventions to be personalized, titrated or adapted [74]. In most interventions, therapists flexibly delivered the intervention adapted to the individual participant (*n* = 8) [79, 85], e.g., including mindfulness in a CBT intervention [78], delivering telephone calls [86], extending the treatment period of CBT, if needed [87], structuring the session of continuation-phase cognitive therapy based on patients' symptoms (C-CT) [83, 84], or a variation in the CBT treatment [89]. To summarise, the most common intervention elements were CBT (*n* = 14), followed by pharmacological treatment (*n* = 7), social support (*n* = 7), and psychoeducation (*n* = 6). The number of elements used in the studies varied between one and nine.

*Implementation strategies.* Implementation strategies, i.e., any enabling or supporting activity [74] were reported in 13 studies [78, 80–86, 89, 94, 96], e.g., training study personnel to deliver CBT as part of the intervention [78, 80, 81, 83–86, 89, 94, 96] (Table 2). Other types of training provided to study personnel included IPT training [78], MCT training [94], and a

five-day training and certification program from the Depression and Bipolar Support Alliance [80]. In most studies, the training sessions included regular supervision from a clinical therapist (psychologist/psychiatrist) with experience in CBT. In other studies, training was delivered as seminars, written materials, or videotapes. Three studies reported that therapists had to complete ≥ 1 year of CBT training and demonstrate competence (scores ≥ 40 on the Cognitive Therapy Scale by Young & Beck [109] before the study treatment. [81, 83, 84]. Tønning et al.'s [79] study reported that a study nurse checked the data three times a week and reacted according to the data presented, providing a double feedback loop between the study nurse and the patient.

Seven studies reported *modifications*, i.e., if the intervention was modified during the study [74]. Overall, modifications were minor and included an extra layer of quality assessment [94]; flexibly delivering CBT [85]; a longer CBT treatment period [89]; and possible medication changes [88] (S9 Appendix).

None of the included studies explicitly reported whether any key contextual elements potentially affected the interventions, e.g., we could not identify any process evaluations, explicit considerations about program theory, or information about contextual elements that potentially shaped interventions during the study period.

**RQ4: What are the findings of the studies as to what promotes recovery from MDD in patients transitioning from outpatient mental health services to primary care?.** *Quantitative findings*. In total, eight quantitative studies [79, 80, 85–87, 90, 94, 96] reported improvement in recovery. The main concepts of these eight interventions were: *MCT versus CBT* [94], *CBT telephone treatment* (CBT-TT) [90], the *MindStep™ program* [86], *group CBT versus individual CBT* [85], *self-management support service* [80], *conventional psychopharmacological, counseling, and/or psychotherapeutic treatment* [87], *a smartphone-based monitoring and treatment system* [79], and *transdiagnostic group CBT versus standard diagnosis-specific group CBT versus* [96].

Studies demonstrating effect on recovery varied according to the delivery method: Face-to-face (*n* = 2), smartphone-app smartphone app with face-to-face feedback (*n* = 1), telephone contacts (*n* = 3) and one study did not report the delivery form of interventions. Likewise, the study duration varied from 6 weeks to 18 months, and most treatment sessions lasted from 30 to 60 minutes. Statistically significant better recovery was reported in Tønning et al.'s RCT study using the RAS scale [79]. Patients in the intervention group were allocated a study nurse who guided and supported patients individually. However, the intensive monitoring and extra contact were deemed stressful by some patients [79]. Another RCT study investigated a self-management system for patients with chronic depressive symptoms using elements of the chronic care model and found significantly higher RAS scores in the intervention group [80]. Two prospective and longitudinal studies followed patients treated in mental health services for either a first episode of MDD [87] or patients with non-chronic MDD [88]. Both studies found that social support was important for patients' recovery process [87, 88] and for obtaining full remission [88]. In both studies, patients received pharmacological treatment, and most patients improved, with 71% of patients having recovered in the 12-month follow-up [87] and 47.1% having recovered six months after the start of their treatment in mental health services [88].

In contrast, six other quantitative studies did not show any effect: *IPT versus CBT* [78], *group CBT* [89], *antidepressant treatment* [88], and *C-CT, clinical management plus FLX, or pill placebo (PBO)* [83, 84]. In addition, Vittengl et al. [81] found that continuing CT could lead to better recovery if patients had responded to acute-phase CT.

*Qualitative findings*. The four qualitative studies and the one mixed method study reported findings from interviews with patients who had previously been treated in mental health services for MDD. In two studies patients were still in outpatient mental health treatment [82,

91]. In one of these studies, patients had received deep brain stimulation (DBS) [82]; they received add-on CBT, and their recovery-process was followed. In this study relatives were also interviewed [82]. Three of the studies explored patients' experiences of specific interventions [82, 86, 91], while in the other two studies patients had not participated in any specific intervention after treatment in mental health services [92, 93]. These latter studies explored patients' conceptions of what had made them recover after treatment in mental health services for MDD.

In the MindStep™ study [86], a structured, guided, low-intensity CBT (LiCBT) was offered after discharge from mental health services in Australia [86]. The treatment was telephone-based and involved an assessment followed by six sessions of about 30 minutes duration. Fourteen patients and four coaches were interviewed. Patients valued coaches' empathy and ability to make rapport, the ease of access by telephone (remote areas in Australia), materials such as self-help guides and workbooks, and that the techniques were individualized to fit the patients' needs. One of the identified barriers was that it required too much homework. The coaches expressed doubt about the sustainability and emphasized that there was a need of better collaboration with mental health care professionals and a clear need to improve transfer and communication with the patient's primary health care professional [86].

In another study [91] patients' experiences of participating in two different forms of occupational therapy groups were explored, one group facilitating return to work and the other focusing on coping with mental illness. All interviewed patients were treated for MDD in outpatient mental health services. Patients appreciated getting their symptoms legitimized by hearing about others with similar symptoms, and they considered workbooks useful. They became aware that life after a depression was different and realized that they had to care for themselves. However, not all patients felt that their problems fit into what was the focus within the groups [91].

The DBS study [82] is the only qualitative study that investigates the experiences of the recovery process following a neurostimulation treatment. It explores patients' and family members' experiences of the process of personal recovery in patients with treatment-resistant depression who had undergone DBS and add-on CBT. Except for a few participants the recovery process was experienced as positive, however very individual. The decisive factor for promoting personal recovery, which implied a reconstructed self after DBS was connectedness and positive relationships with family and others. These relationships were instrumental for patients to build self-confidence, hope, and improve quality of life. Personal recovery might also be achieved without clinical recovery [82].

The two studies [92, 93] without specific interventions explored what patients (women) perceived had made them recover after a hospitalization for major depression. In one study [92] the women found that nursing interventions that could instil hope were useful, as was psychoeducation, reading material, and support groups. However, treatment had to be individualized with varied treatment offers. This study also mentioned patients' need to care for themselves [92]. In the other study [93] the participants mentioned meetings with health care professionals as helpful, but only if the relationship was good and getting access to the health care professional was easy. Individualization and available treatment alternatives were also emphasized here. In addition, a meaningful occupation was considered helpful. The participating women also described the new identity after depression and the need to reflect on their lives to get an enhanced understanding of themselves [93].

There are some common themes in the qualitative studies, except for the DBS study [82] where the focus is narrower. They all mentioned the need to be met as an individual and not just a case and the need to have alternative treatment offers that fit into their specific social context. In addition, the meetings with mental health care professionals were valued, but there

had to be a good relationship. The patients also appreciated reading materials, workbooks, psychoeducation, and support groups. After an episode of MDD, patients experienced a need to reflect on their lives and their new situation and to care for themselves. One study [86] also mentioned the need to improve transfer and communication with the person's primary health care professional. The DBS study [82] focused on how patients reconstructed their sense of self, and this demanded a positive relationship to family and other supporting network.

*Acceptability and fidelity of interventions.* Only a few of the included studies explicitly reported the acceptability of the intervention regarding how it was received by participants and/or met the needs of participants and organizational setting. However, several studies reported dropouts. One study [90] evaluated acceptability using a Likert-type scale to assess satisfaction with telephone counselling. In this study feedback from participants during the 6-month follow-up assessment telephone calls indicated that privacy (e.g., intervention was delivered in their own home) and flexibility of when to receive phone counselling (e.g., phone sessions delivered evening and weekend) were important factors of treatment satisfaction (69% of participants were "very satisfied" with the 8-session CBT-TT) [90].

## Discussion

### Summary of evidence

We aimed to systematically scope, map, and identify the evidence and knowledge gaps on interventions that promote recovery from MDD for patients transitioning from outpatient mental health services to primary care. We found 18 studies with a broad range of heterogeneity in terms of study design, methods, sample size, recovery definitions, recovery rating scales, intervention type, delivery form, and duration.

However, we found limited knowledge about the key interest of this scoping review, i.e., studies about patients who had been treated in outpatient mental health services and were thereafter in a sector-transitioning phase. The studies investigated interventions either for patients in outpatient mental health services before discharge, or for patients after discharge, not specifically targeted to patients in the sector-transitioning phase. The studies that were closest to the focus of the scoping review were the mixed method study by Lawn et al.'s [86], addressing the transition from acute to community mental health care in Australia, and Tønning et al.'s [79] with patients monitoring their mood and health with clinical feedback from a psychiatric nurse after discharge from mental health services. Thus, several studies implemented interventions for patients during their treatment in outpatient mental health services, and two studies [79, 86] implemented interventions immediately after patients' discharge from acute or inpatient mental health services.

The studies used a broad range of recovery rating scales and cut-off scores. However, the concept of recovery was only defined in three studies. Recovery includes both clinical and personal recovery, but most studies only dealt with clinical recovery, although a reduction of symptoms is not automatically an indicator of personal recovery [46, 110].

The most common interventions in the studies were some forms of CBT, followed by pharmacological treatment, social support from by health care professionals and/or family and friends, and psychoeducation. No studies used psychodynamic psychotherapy. The studies that used psychotherapy all used elements of CBT. Overall, the interventions were predominantly delivered to groups or by telephone. All interventions contained several elements and should be categorised as complex interventions [111].

## Eight quantitative studies reported improvement of recovery

The main elements of these studies were CBT, however, also containing other elements. The interventions were delivered face-to-face, by telephone or via a smartphone-app with feedback over telephone. The length and duration of the interventions varied, and they were delivered by different health care professionals, mostly psychologists.

The qualitative studies, except for the DBS study, that had a narrower focus, all mentioned the participants' need to be seen as individuals and not just a case and the need to have alternative treatment offers that could fit into their specific social context. In addition, the meetings with mental health care professionals were valued, but based on a good relationship with that health care professional. The patients also appreciated reading materials, workbooks, psychoeducation, and attending support groups. After severe depression, patients experienced a need to reflect on their new situation and to care for themselves. One study also mentioned the need to improve transfer and communication with the patients' primary health care professional. Many patients recovered after DBS, but positive relationships with family and other supporting network were important.

We found no studies with a shared or collaborative care approach between mental health services and primary care for patients who had been hospitalized with MDD. Different studies showed effect of collaborative care for patients treated for depression in primary care [27, 30, 112]. However, there is a lack of studies, extending this collaborative care approach to patients in the time-period after discharge from outpatient mental health services to ensure maintenance and follow-up of the treatment and prevent relapse. In a collaborative care study in primary care, specially trained care managers, mostly psychiatric nurses, supervised by hospital-based psychiatrists, have been much valued in primary care [113]. However, the social and professional skills of the care managers seemed critical for integrating collaborative care in the primary health care clinic [114]. A recent study of a person-centred coaching approach and liaison work of collaborative care with less educated coaches in primary care for patients with severe mental illness, however mostly focused on patients with schizophrenia and bipolar disorder, did not show effect [115]. Likewise, the CADET study [116] showed less effects than the recent Danish Collaborative care study [30], probably due to differences in the education levels of the care managers. In the Danish study, care managers were psychiatric nurses with a CBT training of at least one-year's duration. It could therefore by hypothesized that collaborative care between mental health services and primary care for patients with MDD in a period after discharge from outpatient mental health services could improve the patients' recovery, detect imminent relapse, and prevent hospitalization.

## Strengths and limitations

We conducted the review in accordance with the PRISMA-ScR in tandem with the Joanna Briggs Institute's (JBI) framework. Concerning data extraction, we followed the template for intervention description and replication (TIDieR) checklist and guide to ensure a systematic extraction of data. Another strength is the involvement of stakeholders guided by the TRANS-FER guide [71], which promoted integration of different perspectives on the aim, design, and methods of the scoping review. However, the scoping review methodology comes with important limitations. They focus on mapping the breath and range of the literature rather than the depth, i.e., the validity of findings. Therefore, we present an overview of the field rather than an evidence synthesis of the probable effect of various types of recovery interventions. In addition, scoping reviews focus on describing knowledge gaps in the literature rather than contributing with new knowledge. We did not assess study quality or bias, nor provide a systematic assessment of the external validity of the evidence, i.e., a GRADE rating. Instead, we outlined

the key characteristics of the best-available evidence in the area. Another limitation concerns the heterogeneity of the included studies. We found studies that included different study populations, e.g., some participants were included from general practice and others from mental health services. Another limitation is that most literature originated from western countries, especially from the USA, and the results may not be applicable in low-and middle-income countries. Thus, the review lacks diversity as it does not represent the global population.

## Implications for research

We identified several major knowledge gaps in the literature. Specifically, no studies investigated interventions aiming to promote recovery in patients transitioning from outpatient mental health services to primary care. There is a need to study interventions that include collaborative, bridge-building efforts in the transitioning phase from mental health services to primary care for patients with MDD. Such studies should investigate other intervention elements than CBT. Integrating patients', health care professionals', and social workers' perspectives on the development of the interventions is paramount in ensuring that the intervention has a good implementation potential [117, 118]. In addition, we identified a variety of scales to measure recovery, which makes it difficult to measure and define recovery in patients with MDD. Therefore, we recommend that future recovery research will investigate this further.

## Conclusion

In this scoping review, we summarised the existing literature of interventions aiming to promote recovery in patient transitioning from outpatient mental health services to primary care, and we identified several knowledge gaps. The studies reported a broad range of heterogeneity in terms of study design, methods, sample size, recovery definitions, recovery rating scales, intervention type, delivery form, and duration. We identified an absence of studies involving patients, who had been treated in mental health services for MDD and were transitioning to primary care. Most studies investigated CBT interventions. In the qualitative studies, participants pointed out, their need to be seen as an individual and to have alternative treatment offers that fit into their specific social context. There is a lack of studies of bridge-building interventions to promote recovery of patients with MDD in the transitioning phase between outpatient mental health services and primary care. This calls for studies of collaborative care for this patient group.

## Supporting information

**S1 Appendix. List of abbreviations / concepts.**
(DOCX)

**S2 Appendix. Preferred Reporting Items for Systematic reviews and Meta-analysis extension for Scoping Reviews (PRISMA-Scr) checklist.**
(DOCX)

**S3 Appendix. Eligibility criteria.**
(DOCX)

**S4 Appendix. Search strategy, PubMed.**
(DOCX)

**S5 Appendix. Sample demographics.**
(DOCX)

**S6 Appendix. Rating scales used to measure recovery outcomes (*n* = 14).**
(DOCX)

**S7 Appendix. Intervention elements (*n* = 16).**
(DOCX)

**S8 Appendix. Content of interventions (*n* = 16).**
(DOCX)

**S9 Appendix. Implementation strategies and modifications (*n* = 16).**
(DOCX)

# Acknowledgments

The review authors would like to thank Lea Fuglsang (LF) for screening title, abstracts, and full texts of articles for inclusion in the review, as well as data extraction.

# Author Contributions

**Data curation:** Anne Sofie Aggestrup, Line Lund Henriksen.

**Formal analysis:** Anne Sofie Aggestrup, Annette Sofie Davidsen.

**Investigation:** Anne Sofie Aggestrup.

**Methodology:** Anne Sofie Aggestrup.

**Project administration:** Anne Sofie Aggestrup.

**Resources:** Frederik Martiny, Annette Sofie Davidsen, Klaus Martiny.

**Supervision:** Frederik Martiny, Annette Sofie Davidsen, Klaus Martiny.

**Validation:** Frederik Martiny, Annette Sofie Davidsen, Klaus Martiny.

**Writing – original draft:** Anne Sofie Aggestrup.

**Writing – review & editing:** Frederik Martiny, Annette Sofie Davidsen, Klaus Martiny.

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
