## [Decision Letter · Decision Letter 0]

5 Feb 2024

PONE-D-23-41386Interventions Promoting Recovery from Depression for Patients Transitioning from Hospital Psychiatry to Primary Care: A Scoping ReviewPLOS ONE

Dear Dr. Aggestrup,

Thank you for submitting your manuscript to PLOS ONE. After careful consideration, we feel that it has merit but does not fully meet PLOS ONE’s publication criteria as it currently stands. Therefore, we invite you to submit a revised version of the manuscript that addresses the points raised during the review process. Please submit your revised manuscript by Mar 21 2024 11:59PM. If you will need more time than this to complete your revisions, please reply to this message or contact the journal office at plosone@plos.org. Please include the following items when submitting your revised manuscript:A rebuttal letter that responds to each point raised by the academic editor and reviewer(s). You should upload this letter as a separate file labeled 'Response to Reviewers'.A marked-up copy of your manuscript that highlights changes made to the original version. You should upload this as a separate file labeled 'Revised Manuscript with Track Changes'.An unmarked version of your revised paper without tracked changes. You should upload this as a separate file labeled 'Manuscript'.If applicable, we recommend that you deposit your laboratory protocols in protocols.io to enhance the reproducibility of your results. Protocols.io assigns your protocol its own identifier (DOI) so that it can be cited independently in the future. For instructions see: https://journals.plos.org/plosone/s/submission-guidelines#loc-laboratory-protocols. Additionally, PLOS ONE offers an option for publishing peer-reviewed Lab Protocol articles, which describe protocols hosted on protocols.io. Read more information on sharing protocols at https://plos.org/protocols?utm_medium=editorial-email&utm_source=authorletters&utm_campaign=protocols.

We look forward to receiving your revised manuscript.

Kind regards,

Lakshit Jain, MD

Academic Editor

PLOS ONE

Journal Requirements:

5. We note that you have referenced (Young, J. and A.T. Beck, Cognitive therapy scale. Unpublished manuscript, University of Pennsylvania, 1980.) which has currently not yet been accepted for publication. Please remove this from your References and amend this to state in the body of your manuscript: (ie “Cognitive therapy scale.[Unpublished]”) as detailed online in our guide for authors http://journals.plos.org/plosone/s/submission-guidelines#loc-reference-style

6. Please include a copy of Table 6 which you refer to in your text on page 23.

**Additional Editor Comments:**

Please respond to comments by the reviewers on a point by point basis.  ==============================

Reviewers' comments:

Reviewer's Responses to Questions

**Comments to the Author**

1. Does the manuscript adhere to the experimental procedures and analyses described in the Registered Report Protocol?

If the manuscript reports any deviations from the planned experimental procedures and analyses, those must be reasonable and adequately justified.

Reviewer #1: Yes

Reviewer #2: Yes

Reviewer #3: Yes

2. If the manuscript reports exploratory analyses or experimental procedures not outlined in the original Registered Report Protocol, are these reasonable, justified and methodologically sound?

A Registered Report may include valid exploratory analyses not previously outlined in the Registered Report Protocol, as long as they are described as such.

Reviewer #1: Yes

Reviewer #2: Yes

Reviewer #3: Yes

3. Are the conclusions supported by the data and do they address the research question presented in the Registered Report Protocol?

The manuscript must describe a technically sound piece of scientific research with data that supports the conclusions. The conclusions must be drawn appropriately based on the research question(s) outlined in the Registered Report Protocol and on the data presented.

Reviewer #1: Yes

Reviewer #2: Yes

Reviewer #3: Partly

4. Have the authors made all data underlying the findings in their manuscript fully available?

Reviewer #1: Yes

Reviewer #2: Yes

Reviewer #3: Yes

5. Is the manuscript presented in an intelligible fashion and written in standard English?

Reviewer #1: Yes

Reviewer #2: Yes

Reviewer #3: Yes

6. Review Comments to the Author

Please use the space provided to explain your answers to the questions above. (Please upload your review as an attachment if it exceeds 20,000 characters)

Reviewer #1: Dear Dr. Lakshit Jain,

I have had the honor of reviewing the article titled "Interventions Promoting Recovery from Depression for Patients Transitioning from Hospital Psychiatry to Primary Care: A Scoping Review” submitted to PLOS ONE for peer review.

The manuscript addresses a very important topic, specifically an area of void where there are no studies extending the effectiveness of collaborative care approach in patients who received inpatient care for MDD and on discharge to ensure continuity of care and relapse prevention. To address this the authors took a scoping review approach.

Recommendations based on the review:

On line 84: The authors mentioned “To our knowledge, no studies have investigated collaborative care for patients who have been hospitalized with severe MDD”

Recommendation: Please be specific that the authors are referring for patients with MDD transitioning into the community and the studies that lack are referring to the collaborative care on discharge for relapse prevention.

On line 92: The authors claim the meaning of clinical recovery as “comprises full symptom remission” questionable if this is accurate as partial symptom improvement can also be considered as clinical recovery.

Lines from 109-111: Please restructure the sentence as the existing terminology relays inconsistent and confusing message.

“Hospital psychiatry” term have been repeatedly used throughout the manuscript, recommend others to either change the wording or specify hospital psychiatric meaning inpatient psychiatric care.

Additional limitations as disclosed by the authors include the scoping review methodology itself where the focus is on the range of the literature versus the depth or validity of findings, the limitation of scoping review included inability to assisted equality or bias to provide a systematic assessment of the extreme validity of the evidence. As the others identified most of the literature is originated from Western countries which lacks diversity as it is not representing global population.

Reviewer #2: Thanks for giving me the opportunity to review this paper. The sample size of 1915 participants is great. I liked the paper but have some concerns.

1. Since there are no studies discussing interventions promoting recovery from depression for patients transitioning from hospital psychiatry to primary care is the title still relevant?

2. The use of different scales to measure recovery in depression makes it difficult to define recovery in the population that was studied.

3. It would be interesting to know which antidepressant medications recovered patients were on.

5. The intervention e.g. CBT varied in terms of methods of delivery and length of delivery. It would be great to see if there is a specific length of time CBT should be delivered that would be conducive to recovery i.e. coming up with standard guidelines and length of treatment.

6. Psychodynamic psychotherapy is a useful modality to explore underlying trauma, conflicts and childhood adverse experiences that drive depression. It would be great to see if it can be used in conjunction with CBT to facilitate long term recovery and address the causes of depression which may be deep rooted in childhood experiences, trauma and unconscious conflicts. It helps by making the patients aware of these unconscious conflicts and promoting lasting change.

Minor mistakes: Row 416 "Versus" should be removed at the end or maybe it is an incomplete sentence and something needs to be added

Row 971, 972: the reference 143 is in a language other than English

Reviewer #3: Strengths of the manuscript:

-The abstract clearly describes the findings of the study.

-The introduction to the article is well laid out.

-Methods and results are clearly explained. The tables included in the manuscript help to review the studies quickly.

-The discussion and conclusion further elaborate on the study findings and suggestions for future research.

Some recommendations are:

-After reading the manuscript title, “Hospital psychiatry,” my initial impression was that the authors are reviewing studies of patients diagnosed with depression transitioning from in-patient settings (e.g., psychiatric hospitals or general hospitals/medical floor, etc.) to outpatient primary care providers. However, the authors use “Hospital psychiatry”, in the title and throughout the manuscript, interchangeably when referring to outpatient and/or in-patient psychiatric care settings, which seems inaccurate. In the study protocol published in 2023, the authors have used the term “outpatient mental health services” in the title. I would appreciate it if the authors could add brief commentary to clarify.

-The authors have primarily included efficacy/effectiveness trials, discussed intervention outcomes, and some studies on variables that may influence intervention outcomes. There are multiple systematic, meta-, and network analyses published comparing different interventions for depression. I will recommend if the authors could add brief commentary discussing how their manuscript is unique compared to previously published studies. The assertion that this is the “first" manuscript to report such findings seems misleading and inaccurate.

-Line 481: typo “...expect for the DBS”

7. PLOS authors have the option to publish the peer review history of their article (what does this mean?). If published, this will include your full peer review and any attached files.

Reviewer #1: No

Reviewer #2: **Yes: **Jasleen Kaur

Reviewer #3: No

---

## [Author Response · Author response to Decision Letter 0]

8 Mar 2024

Dear Academic Editor Dr. Lakshit Jain

Thank you for your feedback and for providing us with the opportunity for revising our manuscript entitled "Interventions Promoting Recovery from Depression for Patients Transitioning from Hospital Psychiatry to Primary Care: A Scoping Review”. We thank the reviewers for their thorough evaluation of our manuscript. Please find a point-to-point reply to yours and the reviewers’ comments in the three tables below. The changes made to the manuscript are marked with track changes in the manuscript. 

Academic Editor

Comments from Editor Answers from authors

https://journals.plos.org/plosone/s/file?id=wjVg/PLOSOne_

formatting_sample_main_body.pdf and 

https://journals.plos.org/plosone/s/file?id=ba62/PLOSOne_

formatting_sample_title_authors_affiliations.pdf

We have ensured that the manuscript meets PLOS ONE’s style requirements, including those for file naming. 

A clean copy of the edited manuscript (uploaded as the new *manuscript* file)” Thank you for your suggestions. The revised manuscript has been proofread by a colleague from out department, Siri Jonina Egede and her partner Luke James Norman. Luke James Norman is a native English speaker and Siri Jonina Egede has lived and worked in England for several years. 

When you resubmit, please ensure that you provide the correct grant numbers for the awards you received for your study in the ‘Funding Information’ section. We are sorry for the mistake. The correct grant information is listed in the ‘Funding Information’ section. 

4. When completing the data availability statement of the submission form, you indicated that you would make your data available on acceptance. We strongly recommend all authors decide on a data sharing plan before acceptance, as the process can be lengthy and hold up publication timelines. Please note that, though access restrictions are acceptable now, your entire data will need to be made freely accessible if your manuscript is accepted for publication. This policy applies to all data except where public deposition would breach compliance with the protocol approved by your research ethics board. If you are unable to adhere to our open data policy, please kindly revise your statement to explain your reasoning and we will seek the editor's input on an exemption. Please be assured that, once you have provided your new statement, the assessment of your exemption will not hold up the peer review process. We are sorry for the mistake in the availability statement. Our raw data is available and was published at the Open Science Framework (OSF) database in connection with our submission of the article to PLOS ONE. 

Please see line 645 in the manuscript: 

All data generated or analysed during this scoping review is available online: 

https://osf.io/5894b/

Line 161: A protocol was prepared in advance and registered in Open Science Framework (OSF) (https://osf.io/ah3sv), published in the medRxiv server (https://doi.org/10.1101/2022.10.06.22280499) and in PLOS ONE. 

5. We note that you have referenced (Young, J. and A.T. Beck, Cognitive therapy scale. Unpublished manuscript, University of Pennsylvania, 1980.) which has currently not yet been accepted for publication. Please remove this from your References and amend this to state in the body of your manuscript: (i.e., “Cognitive therapy scale.[Unpublished]”) as detailed online in our guide for authors http://journals.plos.org/plosone/s/submission-

guidelines#loc-reference-style

Thank you very much for noticing this. The reference has been published. We have therefore inserted the correct reference in line 917 (reference number 109) in the manuscript. 

6. Please include a copy of Table 6 which you refer to in your text on page 23. Thank you for noticing the missing table 6, p. 23. The manuscript does not include a table 6 but an appendix 9, which we have referred to later in the same section. Therefore, it is a mistake and we have deleted the reference to table 6.

7. Please review your reference list to ensure that it is complete and correct. If you have cited papers that have been retracted, please include the rationale for doing so in the manuscript text or remove these references and replace them with relevant current references. Any changes to the reference list should be mentioned in the rebuttal letter that accompanies your revised manuscript. If you need to cite a retracted article, indicate the article’s retracted status in the References list and include a citation and full reference for the retraction notice. Thank you for the reminder to revise the reference list. We have done that.

Reviewer #1

Comments from reviewer #1 Answers from authors

Dear Dr. Lakshit Jain,

I have had the honor of reviewing the article titled "Interventions Promoting Recovery from Depression for Patients Transitioning from Hospital Psychiatry to Primary Care: A Scoping Review” submitted to PLOS ONE for peer review.

The manuscript addresses a very important topic, specifically an area of void where there are no studies extending the effectiveness of collaborative care approach in patients who received inpatient care for MDD and on discharge to ensure continuity of care and relapse prevention. To address this the authors took a scoping review approach. Thank you very much for your positive response. 

On line 84: The authors mentioned “To our knowledge, no studies have investigated collaborative care for patients who have been hospitalized with severe MDD”

Recommendation: Please be specific that the authors are referring for patients with MDD transitioning into the community and the studies that lack are referring to the collaborative care on discharge for relapse prevention. Thank you for your suggestion, we agree

that we could be more specific. 

We have added a more specific text in the

manuscript in line 88: 

“To our knowledge, no studies have investigated collaborative care for relapse prevention for patients with MDD who have been discharged from outpatient mental health services (i.e., outpatient hospital-based mental health services, S1 Appendix) and continue their treatment in primary care”. 

On line 92: The authors claim the meaning of clinical recovery as “comprises full symptom remission” questionable if this is accurate as partial symptom improvement can also be considered as clinical recovery. We agree that clinical recovery can also be considered and defined by authors as partial symptom improvement. 

We have changed the text in the manuscript accordingly, in line 98: 

“Authors define clinical recovery as comprised partial or full symptom remission, independent living, gaining control over the illness, and full or part-time work or education”. 

Lines from 109-111: Please restructure the sentence as the existing terminology relays inconsistent and confusing message. Thank you for noticing this. We agree that the sentence is unclear.

In the manuscript, in line 115-117, we have

added the following: 

“Research shows that most patients with MDD do not achieve symptomatic remission or full recovery after treatment in outpatient mental health services. Therefore, they are at risk of developing new episodes of MDD or treatment-resistant depression”. 

“Hospital psychiatry” term have been repeatedly used throughout the manuscript, recommend others to either change the wording or specify hospital psychiatric meaning inpatient psychiatric care. Thank you for your comment. In our protocol article, we used the concept "mental health services". Therefore, we have decided to keep the concept from the protocol article. We have changed the concept in both the title of the article and throughout the manuscript from “hospital psychiatry” to “mental health services”. 

However, to make it clear that the transition of patients is from hospital-based outpatient psychiatry to primary care, we have added the concept and its precise definition to S1 Appendix: 

Abbreviation/concept Definition

Outpatient mental health services Outpatient hospital-based mental health services

Additional limitations as disclosed by the authors include the scoping review methodology itself where the focus is on the range of the literature versus the depth or validity of findings, the limitation of scoping review included inability to assisted equality or bias to provide a systematic assessment of the extreme validity of the evidence. As the others identified most of the literature is originated from Western countries which lacks diversity as it is not representing global population. Thank you for sharing your reflection. We agree that it is a limitation. However, we have already written about these limitations in the discussion section (sub-section: “Strengths and limitations”). 

Please see line 592-596 (limitations regarding scoping reviews as a method). 

Please see line 601-604 (limitations regarding Western countries). In addition, we have added a final sentence to this theme in line 603: “Thus, the review lacks diversity as it does not represent the global population”. 

Reviewer #2

Comments from reviewer #2 Answers from authors

Thanks for giving me the opportunity to review this paper. The sample size of 1915 participants are great. I liked the paper but have some concerns. Thank you very much for your positive response. 

1. Since there are no studies discussing interventions promoting recovery from depression for patients transitioning from hospital psychiatry to primary care is the title still relevant? Thank you. We believe that the title is still relevant, as it expresses our aim and the research questions of the review and the perspectives, we investigate. Only few studies dealt with the very transitioning phase. However, we also included studies where patients were being prepared for transition at the end of their outpatient treatment being prepared for transition. 

We have already mentioned this in the discussion section in the manuscript, line 532-536: “However, we found limited knowledge about the key interest of this scoping review, i.e., studies about patients who had been treated in outpatient mental health services and were thereafter in a sector-transitioning phase. The studies investigated interventions either for patients in outpatient mental health services before discharge, or for patients after discharge, not specifically targeted to patients in the sector-transitioning phase”. 

2. The use of different scales to measure recovery in depression makes it difficult to define recovery in the population that was studied. Thank you for your reflection. We agree that that the use of different recovery scales makes it difficult to measure and define recovery. It was one of our findings that there was a broad variation in scales and definitions. We recommend future research to investigate this further. In the section “Implications for research” (p. 32 in the manuscript) we have added in line 614: 

“In addition, we identified a variety of scales to measure recovery, which makes it difficult to measure and define recovery in patients with MDD. Therefore, we recommend that future recovery research will investigate this further”.

3. It would be interesting to know which antidepressant medications recovered patients were on. We agree that it would be interesting to know which anti-depressant medication the recovered patients received. We did not mention the names of the medication in the review, but they can be found in the uploaded material in our online raw-data extraction file. 

4. The intervention e.g., CBT varied in terms of methods of delivery and length of delivery. It would be great to see if there is a specific length of time CBT should be delivered that would be conducive to recovery i.e., coming up with standard guidelines and length of treatment. Thank you for your comment. Information on methods of delivery and length of delivery is included in S8 Appendix. However, the influence of different forms of delivery cannot be examined in a scoping review. 

5. Psychodynamic psychotherapy is a useful modality to explore underlying trauma, conflicts and childhood adverse experiences that drive depression. It would be great to see if it can be used in conjunction with CBT to facilitate long term recovery and address the causes of depression which may be deep rooted in childhood experiences, trauma, and unconscious conflicts. It helps by making the patients aware of these unconscious conflicts and promoting lasting change. Thank you. We have added the following on line 549-552 in the manuscript: 

“No studies used psychodynamic psychotherapy. Those studies that used psychotherapy all used elements of CBT. Overall, the interventions were predominantly delivered to groups or by telephone”. 

Minor mistakes: Row 416 "Versus" should be removed at the end or maybe it is an incomplete sentence, and something needs to be added

Row 971, 972: the reference 143 is in a language other than English Thank you for noticing these mistakes. 

We have removed the “versus” at the end of the sentence (line 416). 

Regarding reference 143 (line 998): The Abstract is both in France and in English (https://www.sciencedirect.com/science/article

/abs/pii/S0013700614001109). 

The reference is mentioned in S8 Appendix: Content of interventions. The reference is included in one of our included studies from the literature search (Wooley et al. 2020). Wooley et al. (2020) describe that their material used in their intervention was inspired by: 

“The Recovery Workbook (143) was used in the LMI group, and the cognitive behavioural therapy intervention for individuals in supported employment (144) in the RTW group”.

Please see S8 Appendix. 

Reviewer #3

Comments from reviewer #3 Answers from authors

Strengths of the manuscript:

-The abstract clearly describes the findings of the study.

-The introduction to the article is well laid out.

-Methods and results are clearly explained. The tables included in the manuscript help to review the studies quickly.

-The discussion and conclusion further elaborate on the study findings and suggestions for future research. Thank you very much for your positive evaluation of the strengths of the manuscript. 

Some recommendations are:

-After reading the manuscript title, “Hospital psychiatry,” my initial impression was that the authors are reviewing studies of patients diagnosed with depression transitioning from in-patient settings (e.g., psychiatric hospitals or general hospitals/medical flo

---

## [Decision Letter · Decision Letter 1]

1 Apr 2024

Interventions Promoting Recovery from Depression for Patients Transitioning from Outpatient Mental Health Services to Primary Care: A Scoping Review

PONE-D-23-41386R1

Dear Dr. Aggestrup,

We’re pleased to inform you that your manuscript has been judged scientifically suitable for publication and will be formally accepted for publication once it meets all outstanding technical requirements.

Kind regards,

Lakshit Jain, MD

Academic Editor

PLOS ONE

Additional Editor Comments (optional):

Reviewers' comments:

Reviewer's Responses to Questions

**Comments to the Author**

1. Does the manuscript adhere to the experimental procedures and analyses described in the Registered Report Protocol?

If the manuscript reports any deviations from the planned experimental procedures and analyses, those must be reasonable and adequately justified.

Reviewer #1: Yes

Reviewer #3: Yes

2. If the manuscript reports exploratory analyses or experimental procedures not outlined in the original Registered Report Protocol, are these reasonable, justified and methodologically sound?

A Registered Report may include valid exploratory analyses not previously outlined in the Registered Report Protocol, as long as they are described as such.

Reviewer #1: Yes

Reviewer #3: Yes

3. Are the conclusions supported by the data and do they address the research question presented in the Registered Report Protocol?

The manuscript must describe a technically sound piece of scientific research with data that supports the conclusions. The conclusions must be drawn appropriately based on the research question(s) outlined in the Registered Report Protocol and on the data presented.

Reviewer #1: Yes

Reviewer #3: Yes

4. Have the authors made all data underlying the findings in their manuscript fully available?

Reviewer #1: Yes

Reviewer #3: Yes

5. Is the manuscript presented in an intelligible fashion and written in standard English?

Reviewer #1: Yes

Reviewer #3: Yes

6. Review Comments to the Author

Please use the space provided to explain your answers to the questions above. (Please upload your review as an attachment if it exceeds 20,000 characters)

Reviewer #1: Thank you for addressing the comments for the manuscript titled "Interventions Promoting Recovery from Depression for Patients Transitioning from Outpatient Mental Health Services to Primary Care: A Scoping Review” submitted to PLOS ONE for peer review.

All the comments were appropriately addressed, and corresponding changes reflect the reviewers’ recommendations.

Reviewer #3: The authors have adequately answered the questions raised by the reviewers, and it is now suitable for publication.

7. PLOS authors have the option to publish the peer review history of their article (what does this mean?). If published, this will include your full peer review and any attached files.

Reviewer #1: No

Reviewer #3: No

---

## [Editor Report · Acceptance letter]

25 Apr 2024

PONE-D-23-41386R1 

PLOS ONE

Dear Dr. Aggestrup, 

I'm pleased to inform you that your manuscript has been deemed suitable for publication in PLOS ONE. Congratulations! Your manuscript is now being handed over to our production team.

Kind regards, 

on behalf of

Dr. Lakshit Jain 

Academic Editor

PLOS ONE